# Inorganic Polyphosphate Modulates Chromosome Transmission Fidelity in the Fission Yeast *Schizosaccharomyces pombe*

**DOI:** 10.3390/biom15091331

**Published:** 2025-09-18

**Authors:** Sarune Bollé, Elisa Koc, Adolfo Saiardi, Lisa Juhran, Eva Walla, Ursula Fleig, Abel Alcázar-Román

**Affiliations:** 1Eukaryotic Microbiology, Department of Biology, Heinrich-Heine-University, Universitätsstrasse 1, 40225 Düsseldorf, Germany; sadau101@hhu.de (S.B.); elkoc100@hhu.de (E.K.); lijuh100@hhu.de (L.J.); eva.walla@hhu.de (E.W.); 2Medical Research Council Laboratory for Molecular Cell Biology, University College London, Gower St., London WC1E 6BT, UK; a.saiardi@ucl.ac.uk

**Keywords:** inorganic polyphosphate, polyP, genome stability, kinetochore, CCAN complex, gene dosage, aneuploidy, *Schizosaccharomyces pombe*, yeast

## Abstract

Chromosome transmission fidelity is vital for organism fitness. Yet, extrinsic and intrinsic changes can affect this process, leading to aneuploidy, the loss/gain of chromosomes, which is a hallmark of cancer. Here, using a haploid fission yeast *Schizosaccharomyces pombe* strain with a segmental aneuploidy, we assayed genome stability under different temperatures and altered gene dosage. We find that *S. pombe* genome stability is temperature-dependent and is unexpectedly modulated by intracellular levels of inorganic polyphosphate polymers (polyP). The *vtc4^+^* gene, encoding a subunit of the polyP-generating VTC complex, is present twice due to the segmental aneuploidy resulting in a gene-dosage-coupled increase in polyP. Using strains with different amounts of polyP, we find a direct negative correlation between polyP and chromosome segregation fidelity. PolyP modulates the function of the conserved CCAN kinetochore subcomplex, as the abnormal growth phenotype caused by the mutant CCAN protein Fta2-291 was rescued in the absence of polyP, while extra polyP had the opposite effect. Importantly, this appears to occur in part by modulation of the nucleolin Gar2. Gar2 is the functional homolog of the *Saccharomyces cerevisiae* Nsr1 protein, whose function is modulated by posttranslational polyP-mediated polyphosphorylation. Thus, polyP modulates genome stability, linking cellular metabolism to chromosome transmission fidelity.

## 1. Introduction

When a cell divides, the duplicated sister chromosomes need to be segregated to each of the two daughter cells by the mitotic machinery for the attainment of genome stability. Errors in this process can result in aneuploidy: the gain or loss of chromosomes. An uneven number of chromosomes usually has a negative impact on the fitness of a cell/organism, and this has been linked to various human genetic disorders and is found in approximately 90% of solid tumors [1,2]. Analysis of laboratory strains of the budding yeast *Saccharomyces cerevisiae* has shown that disomic strains have a lower proliferation rate and that in general, the severity of the phenotype is linked to the gene number/size of the extra chromosome [3]. At the cellular level, the fitness of aneuploid cells is decreased by a number of stress factors. These can be caused indirectly or directly such as the occurrence of specific gene expression patterns or gene dosage imbalance [4,5,6]. Aneuploidy can contribute to genomic instability per se, but there are also examples in both human and yeast cells, where a specific gene present on the disomic chromosome increases the chromosome instability (CIN) of the aneuploid cell [4,7]. Human trisomy 13 is linked to increased chromosomal mis-segregation due to the presence of the *SPG20* gene on chromosome 13, while chromosome VI disomy lethality in *S. cerevisiae* is partially attributed to the presence of the extra copy of TUB2, encoding β-tubulin [8,9]. Despite the documented negative impact caused by aneuploidy or segmental aneuploidy in various systems, aneuploidy is prevalent in *S. cerevisiae* isolates from highly diverse origins specifically also in natural, non-domesticated isolates [10]. Furthermore, under specific selection pressures, aneuploid populations derived from *S. cerevisiae* laboratory strains have an increased fitness compared to the wild-type ancestor and pathogenic fungi such as *Candida albicans* that use aneuploidy to survive under extreme stress conditions [11,12]. Thus, aneuploidy can benefit an organism under specific circumstances.

In contrast to yeasts like *S. cerevisiae*, which exist in haploid and diploid states, fission yeasts such as *Schizosaccharomyces pombe* (*S. pombe*) have a haplontic life cycle, and the diploid phase is restricted to zygotes. Laboratory-generated diploid *S. pombe* cells are more sensitive to genotoxic stress and show increased genome instability [13,14]. Only a few stable aneuploid strains exist in *S. pombe*, and these are disomic haploids, which carry the linear Ch16 mini-chromosome (Ch16 MC) or shorter variants thereof. The Ch16 MC strain, which was used in the present study, is a stable, segmental aneuploid strain with three endogenous chromosomes and a 530 kb, non-essential mini-chromosome (MC) named Ch16 derived from the 3.5 Mb chromosome III [15,16,17]. The Ch16 MC contains 163 complete ORFs also present on chromosome III plus centromere and telomere sequences [18]. One of the genes present both on chromosome III and the Ch16 MC is *vtc4^+^*, encoding the polyphosphate synthase [19]. The average of the expression levels of the majority of the 163 ORFs was 1.9 in the partial aneuploid strain, demonstrating that expression increases in proportion to the gene copy number, and dosage compensation does not occur [7,18]. Presence or absence and thus loss of the non-essential MC can be scored by a colony color assay, and this assay has been used extensively to identify the proteins required for chromosome segregation fidelity [20,21,22,23,24,25].

In the present study, we employed Ch16 MC-carrying strains to determine the effects of (i) extrinsic and (ii) intrinsic changes on the transmission fidelity of the extra chromosome. We chose variations in temperature as the environmental challenge and alteration of gene dosage as an internal variation. For temperature alteration, we did not use extreme hot or cold temperature stress, as such environmental conditions affect the cell’s cytoskeleton and thus are likely to have an impact on mitotic spindle microtubules, but used temperatures between 25 and 36 °C [26]. For gene dosage alterations, we used the *vtc4^+^* gene, which is one of the 163 genes present both on chromosome III and the Ch16 MC. Vtc4 is the catalytic subunit of the vacuole transporter chaperone (VTC) complex that is responsible for inorganic polyphosphate (polyP) synthesis [19,27,28]. PolyP comprises a linear polymer of phosphate residues, present in all living organisms and regulating a rapidly growing number of biological processes, as well as being associated with a number of human diseases [29,30,31,32,33,34,35,36]. These include modulation of phosphate and cation homeostasis, energy metabolism, stress response, cell cycle progression and protein modification via lysine polyphosphorylation [37,38,39,40,41]. To date, polyP has not been implicated in chromosome transmission fidelity in any system. However, we have shown previously that in *S. pombe*, chromosome segregation fidelity and polyP synthesis are both regulated by the enzymatic activity of Asp1, a member of the highly conserved PPIP5K family [19,42,43,44]. Asp1 is a bifunctional enzyme that generates inositol pyrophosphates, which affect spindle function and localization of kinetochore proteins belonging to the conserved CCAN kinetochore complex in a dose-dependent manner [19,42,43,44]. We find that both temperature and the presence of polyP have a significant impact on the chromosome transmission fidelity of the Ch16 MC, and we show that polyP affects kinetochore function in a dose-dependent manner.

## 2. Materials and Methods

### 2.1. Strains and Plasmids

Strains, plasmids, and oligonucleotides used in this study are listed in Appendix A. A strain with a deletion of the *vtc4^+^* open reading frame (ORF) on the Ch16 mini-chromosome (MC) was generated by replacing the ORF with a *TEF: hygR* cassette via PCR-based homologous recombination [45]. Transformation was performed as described in [46], with selection of transformants on plates containing hygromycin (100 µg/mL; Roth). Colonies retaining the Ch16 MC are white and were identified by growth on adenine-deficient medium. Successful deletion of the *vtc4^+^* ORF on the Ch16 MC was confirmed by diagnostic PCR and differential growth of red (Ch16 MC lost) and white (Ch16 MC retained) colonies on hygromycin plates. All other strains were generated through genetic crosses followed by tetrad dissection or random spore analysis. All plasmids were constructed according to established protocols [44,47]. The pJR2-3XL vector containing the thiamine-repressible *nmt1^+^* promoter [48] was used for *vtc4^+^* overexpression studies. Gene sequences were obtained from PomBase.

### 2.2. Media and Growth Conditions

*S. pombe* strains were usually cultured at 30 °C except kinetochore mutant strains which were pregrown at 25 °C. Strains were grown in rich medium (YE5S with 3% glucose) or minimal medium (MM with 2% glucose and supplements) [49]. Cultures were shaken at 150 rpm or grown on agar plates (2% agar). For gene expression via the *nmt1^+^* promoter, cells were cultured in MM supplemented with 5 µg/mL thiamine (low expression) or without thiamine (high expression). For serial dilution patch assays of the temperature-sensitive kinetochore mutants, transformants were pre-cultured in liquid MM (-leu, +thiamine) at their respective permissive temperatures (25 °C for most strains; 28 °C for *gar2Δ*). Logarithmically growing cells (10^4^ to 10^1^) were spotted onto MM -leu (+ and − thiamine) agar plates and were incubated under the control and restrictive conditions, as indicated in the figures and corresponding legends. Yeast strains which were tested on full media YE5S plates were precultured in liquid YE5S and then spotted onto YE5S plates.

### 2.3. Mini-Chromosome Loss Assay

Presence or absence of the Ch16 MC [16] was assessed by plating ~2000 cells/plate from mid-log phase cultures (OD_600_ = 0.5–1.0), grown in adenine-deficient minimal medium to prevent loss of the Ch16 MC, onto medium containing limiting adenine (5 µg/mL). Plates were incubated for 7 days at 25 °C, 30 °C, or 36 °C. To enhance colony coloration, plates were stored at 4 °C for 24 h before scoring. White (Ch16 MC present) and sectored white/red colonies (Ch16 MC lost in some cells of the colony) were counted. Entirely red colonies were not counted, as these cells had lost the Ch16 MC before plating.

### 2.4. Isolation of Nuclei

Nuclei were isolated from *S. pombe* as previously described [50]. Briefly, cells were grown in YE5S at 30 °C to OD_600_ = 1, harvested by centrifugation (4000× *g*, 5 min), and washed twice with Buffer S (1.4 M sorbitol, 40 mM HEPES pH 6.5, 0.5 mM MgCl_2_). Cells were resuspended in Buffer S supplemented with 10 mM β-mercaptoethanol and 1 mM PMSF (Sigma-Aldrich, St. Louis, MO, USA), incubated at 30 °C for 10 min, and spheroplasted using 100T Zymolyase (20 mg/g; Roth, Carl Roth GmbH Karlsruhe, Germany). After lysis in 1% SDS on ice, lysates were washed and resuspended in F buffer (18% Ficoll 400, 20 mM PIPES pH 6.5, 0.5 mM MgCl_2_) containing 1 mM PMSF and protease inhibitor mix (cOmplete™, Roche, Basel, Switzerland) and homogenized using a Dounce homogenizer (10 strokes with a tight-fitting pestle). The homogenate was layered onto GF buffer (7% Ficoll 400, 20% glycerol, 20 mM PIPES pH 6.5, 0.5 mM MgCl_2_) and centrifuged at 20,000× *g* for 30 min at 4 °C. The organelle-enriched pellet was washed and re-centrifuged; the resulting nuclear pellet was resuspended in F buffer for polyphosphate quantification.

### 2.5. Polyphosphate Extraction and Analysis

Polyphosphate (polyP) was extracted as described [19,51] with minor modifications. Exponentially growing cells (OD_600_ = 0.5; 20 OD units) were lysed in LETS buffer (0.1 M LiCl, 10 mM EDTA, 0.5% SDS, 10 mM Tris-HCl pH 8.0) supplemented with acidified phenol (pH 4.8; Sigma-Aldrich). Cell disruption with glass beads was performed using a Precellys 24 homogenizer (2 × 20 s at 4000 U). The aqueous RNA/polyP phase was recovered by phenol–chloroform extraction, precipitated with ethanol (−20 °C, overnight), dried at 65 °C, and resuspended in nuclease-free water. RNA concentration was determined spectrophotometrically with a NanoDrop (2000 c, Peqlab, Erlangen, Germany).

For qualitative analysis, 20 µg RNA per sample was resolved by native PAGE (35.5% acrylamide: bis-acrylamide, 19:1; 1× TBE) at 4 °C for 20 h and stained with toluidine blue (Sigma-Aldrich) or DAPI, as indicated in the figure legends. For quantitative polyP analysis, samples were digested with recombinant *S. cerevisiae* Ddp1 and Ppx1 (expressed in *E. coli* Rosetta DE3). Reactions were performed at 37 °C for 1 h in 5 times reaction buffer (150 mM HEPES pH 6.8, 250 mM NaCl, 30 mM MgSO_4_, 5 mM DTT). Released phosphate was quantified by a malachite green assay (4:3 ratio of reaction mixture to freshly prepared molybdate/malachite green reagent; Roth), with absorbance measured at 640 nm. Pi concentration was calculated using a KH_2_PO_4_ standard curve. For nuclear polyP extraction, isolated nuclei were lysed in LETS buffer (0.1 M LiCl, 10 mM EDTA, 10 mM Tris-HCl pH 7.4, 0.2% SDS), mechanically sheared by repeatedly passing the lysate through a 0.2 µm needle and subsequently extracted with acidic phenol (pH 4.3) as described above. Nuclear polyP quantification: Cells were grown in 1 L YE5S to OD_600_ = 1. From this culture, 20 mL (20 OD units) were harvested for polyP extraction from whole cell lysates (WCL). The remaining 980 OD units were used for nuclear fractionation, yielding three nuclear pellets. One pellet (327 OD units) was used for polyP extraction. For polyP PAGE analysis, 2.6 OD units of WCL and 327 OD units of nuclear extract were loaded per lane, corresponding to an input ratio of 1:126. polyP bands were quantified directly from the gel using ImageJ, and values were normalized to the input OD ratio.

### 2.6. Western Blot Analysis

Nuclear pellets were resuspended in 1× NuPAGE™ LDS sample buffer (Invitrogen™, Waltham, MA, USA, supplemented with 10% β-mercaptoethanol). Whole cell lysates from *S. pombe* were prepared as previously described [39]. In brief, 50 OD units of logarithmically growing cells were lysed using glass beads and subsequently resuspended in 1× LDS sample buffer. Equal volumes (20 µL) of nuclear and whole cell lysates were loaded onto Bis-Tris gels (Invitrogen™) and transferred to PVDF membranes using a semi-dry blotting system. Successful transfer was verified by staining the membranes with 0.1% Ponceau S in 5% acetic acid. Blocking was performed 1 h at RT in 3% milk, TBS-T. Western blotting was performed using the following primary antibodies: polyclonal anti-Histone H3 (rabbit, 1:1000; Cell Signaling Technology #9715, Danvers, MA, USA) and monoclonal anti-GAPDH (mouse, 1:1000; Sigma-Aldrich #G8795). Secondary antibodies were HRP-conjugated goat anti-rabbit IgG (1:2500; Thermo Fisher Scientific #31460) and goat anti-mouse IgG (1:2500; Thermo Fisher Scientific #31430). Protein bands were visualized using the Pierce™ ECL Western Blotting Substrate (Invitrogen™) and a ChemiDoc Imaging System (Bio-rad, Hercules, CA, USA).

### 2.7. Microscopy

Mitotic defects were analyzed by shifting temperature-sensitive, logarithmically growing (liquid YE5S) *fta2-291 and fta2-291, vtc4Δ* strains from 25 °C to 33 °C for 6 h, followed by fixation with 70% ethanol and staining with DAPI (0.1 µg/mL; Sigma-Aldrich, #D9542). Nuclear segregation phenotypes were categorized as equal (wild-type) or unequal (mutant) based on the DNA signal distribution of mitotic cells. For live-cell imaging of Gar2-GFP or Gar2-mCherry cells, cells were grown at 30 °C in liquid or solid minimal medium with appropriate supplements, mounted on 2% agarose pads prepared in the same medium and sealed with VALAP (1:1:1 vaseline: paraffin: lanolin). Imaging was performed using a spinning disk confocal microscope (Olympus, Tokyo, Japan) equipped with a 100× oil immersion objective. To quantify the mitochondrial distribution of Cox4-RFP, a rectangular region of interest (ROI) was drawn along the longitudinal axis of each cell using ImageJ. The Plot Profile function was applied to this ROI to measure the average gray value per pixel column, generating an intensity profile along the cell’s long axis. In this case, a LSM 880 Airyscan (Zeiss, Oberkochen, Germany) was used with a 63× immersion oil objective. For imaging and quantification of Fta2-GFP levels at kinetochores, cells were grown at 25 °C and processed for microscopy as described above. Quantification of signals was performed using a macro-based fluorescence analysis. A MIP was created and regions of interest (kinetochores) defined using the following macro [44]:

run(“Duplicate…”, “ “);

run(“8-bit”);

run(“Subtract Background…”, “rolling = 50”);

run(“Maximum…”, “radius = 2”);

run(“Threshold…”);

waitForUser(“thr”, “setzit”);

setOption(“BlackBackground”, false);

run(“Convert to Mask”);

run(“Analyze Particles…”, “size = 0.20–2.00 circularity = 0.80–1.00 display exclude clear include

add”);

A threshold was set for all experimental sets so that most GFP signals were defined as a region of interest (ROI) while as little artifacts as possible were marked. Subsequently, ROIs were manually curated to make sure only kinetochores were identified. The signal intensity in the ROIs was measured by transferring the saved ROI-set to the unedited MIP and using the measure option. The value used for quantification was the Integrated Density (IntDen) value as it accounts for the area and the mean gray value of the ROI. For the adjusted integrated density value, the background signal in an area equal to that of the ROI was subtracted from each value. The number of quantified signals is given as the number of kinetochore signals in the figure legends.

### 2.8. Programs and Statistical Analysis

Image analysis was performed using ImageJ version 1.54p (NIH) with the Olympus Viewer Plugin. Quantification of Fta2-GFP fluorescence was performed using a custom macro written in the ImageJ macro language (see Appendix A). Statistical analysis and graphing were performed with GraphPad Prism v10.4.1.

## 3. Results

### 3.1. Chromosome Transmission Fidelity Is Temperature Dependent

Temperature has a wide impact on biological processes, especially at more extreme temperatures [52]. We wanted to determine whether moderate temperature changes affected chromosome transmission fidelity and thus quantified loss of the Ch16 MC at growth temperatures of 25, 30 and 36 °C. Presence or absence of the non-essential MC can be determined on solid media by a colony color assay of cells which carry two *ade6* alleles; *ade6-M210* on chromosome III and *ade6-M216* on the MC (Figure 1a). Presence of both *ade6* alleles results in intragenic complementation, adenine prototrophy and white colonies, while loss of the MC results in adenine auxotrophy and red colonies on indicator plates. Red sectors in a white colony (sectored colony) served as a readout for Ch16 MC loss. (Figure 1b). Cultures of Ch16 MC harboring cells were grown in liquid under Ch16 MC selective conditions and plated on solid media under Ch16 MC non-selective conditions (indicator plates) at 25 °C, 30 °C and 36 °C. Sectoring colonies were counted and quantified as percentage of total colonies. We found that chromosome transmission fidelity was reduced significantly with increasing temperature. The number of sectoring colonies was lowest at an incubation temperature of 25 °C and increased 1.8-fold at an incubation temperature of 30 °C and 9-fold at an incubation temperature of 36 °C compared to plates incubated at 25 °C (Figure 1c–d). Thus, increasing temperature decreases chromosome transmission fidelity, but the decrease is not linear and highly pronounced at 36 °C incubation temperature.

### 3.2. vtc4^+^ Gene Dosage Impacts polyP Levels

Our results show that a strain harboring the Ch16 MC shows a temperature-dependent chromosome transmission instability. We next asked whether this might be due to the partial disomy of chromosome III and thus be caused by a gene dosage effect. Analysis of the ORFs present both on chromosome III and the Ch16 MC revealed that the *vtc4^+^* gene is present in chromosome III and the long arm of the Ch16 MC, directly downstream to the *ade6^+^* locus (Figure 2a).

The Vtc4 protein is part of the vacuole transporter chaperone (VTC) complex responsible for inorganic polyphosphate (polyP) synthesis and comprises the catalytic domain. We had shown previously that polyP synthesis is regulated by the *S. pombe* PPIP5K family member Asp1, a bifunctional enzyme with kinase and pyrophosphatase activity responsible for establishing the levels of 1-IPPs (=1-IP_7_ and 1,5-IP_8_) in the cell (Figure 2b) [19,55,56]. 1,5-IP_8_ interacts with the SPX domains of the *S. cerevisiae* VTC complex components Vtc4 and Vtc2, promoting the active conformation of the complex, which stimulates polyP polymerization. In *S. pombe*, higher than wild-type levels of 1,5-IP_8_ increase polyP synthesis, whereas depletion of 1,5-IP_8_ greatly diminishes its production [19]. As Asp1 enzymatic activity modulates polyP synthesis and has been shown to be an important contributor to chromosome segregation fidelity via modulating spindle dynamics and the kinetochore complex [42,44] (Figure 2b), we asked whether (i) two copies of *vtc4^+^* result in increased polyP generation and (ii) whether such an increase might affect transmission fidelity of the Ch16 MC.

To test whether harboring an extra *vtc4^+^* copy had any impact on polyP levels, a haploid wild-type strain and the Ch16 MC strain were grown in rich liquid media, and polyP was extracted from logarithmically growing cultures followed by PAGE analysis of phenol extracted polyP. We observed that the strain harboring the Ch16 MC had approximately 37% more polyP when compared to a wild-type control (Figure 2c). Next, to investigate the contribution of each copy of *vtc4^+^* in our Ch16 MC-harboring cells on polyP generation, we constructed strains with deletions of *vtc4^+^* present on the endogenous ChIII, on Ch16 MC, or on both. These strains all showed similar growth behavior (Appendix A). Compared to the strain with the two *vtc4^+^* ORFs, the strains with only one copy of *vtc4^+^* showed ~30% reduction in polyP levels: 34% reduction for the deletion in ChIII and 28% reduction for the deletion in Ch16. The Ch16MC strain where both copies of *vtc4^+^* had been deleted had no polyP (Figure 2d and quantification in Figure 2e).

Thus, the presence of two chromosomal copies of *vtc4^+^* due to the presence of the Ch16 MC results in a 30% increase in total cellular polyP levels in a Ch16 MC-harboring strain even though the genes encoding the other members of the VTC complex are only present in one copy in this strain (*vtc1^+^* in chromosome II; *vtc2^+^* in chromosome I) [57]. If polyP has an impact on chromosome transmission fidelity, one would expect that it acts in the nucleus, especially as *S. pombe* has a closed mitosis with no breakdown of the nuclear membrane.

The bulk of the cellular polyP is concentrated in the vacuole of yeast cells, but polyP has also been identified in other organelles, albeit at lower levels [39,58,59]. To investigate the possible existence of nuclear polyP, we performed polyP extractions from nuclear fractions of logarithmically growing strains that generated wild-type, higher-than-wild-type and no polyP. This was achieved by using a wild-type strain, the Asp1-variant Asp1^H397A^-expressing strain and a *vtc4^+^* deletion strain, respectively [19,60]. As previously described, higher than wild-type levels of polyP were found in whole cell lysates (WCL) of the *asp1^H397A^* strain and nearly undetectable in the *vtc4Δ* strain (Figure 2f) [19]. Importantly, polyP was also detected in nuclear lysates of wild-type and *asp1^H397A^* strains (Figure 2f). The polyP in the nuclear fraction was approximately 0.25–0.3% (for *asp1^H397A^* and wild-type strain, respectively) of the WCL (calculation found in the materials section) (Figure 2f and Appendix A). Interestingly, staining was strongest in the higher molecular weight area of nuclear samples when compared to WCL (Figure 2f).

### 3.3. Cellular polyP Levels Impact Chromosome Transmission Fidelity in a Dose-Dependent Manner

To investigate whether the increased polyP levels of the Ch16 MC strain with two *vtc4^+^* copies had an impact on chromosome transmission fidelity, we quantified sectoring colonies of strains with deletions of *vtc4^+^* in either the endogenous ChIII, in Ch16 MC, or in both at 30 °C. Interestingly, we observed that compared to the strain harboring 2 copies of *vtc4^+^*, strains containing either 1 or no copies of *vtc4^+^* showed significantly lower levels of Ch16 MC loss, as well as a 31–36% and 61% reduction in the median, respectively (Figure 3a). Thus, there is a dose-dependent impact of polyP on chromosome transmission fidelity. The genomic location of *vtc4^+^* had no impact on chromosome transmission as MC strains containing only one copy of *vtc4^+^*, on either Ch16 MC or ChIII, which showed a similar loss of the MC (Figure 3a). The strain harboring no copies of *vtc4^+^* showed a 39–44% reduction in the median chromosome loss when compared to a strain with one copy of *vtc4^+^* in either Ch16 (*p* < 0.05) or ChIII (*p* = 0.052) (Figure 3a).

Next, we determined whether the highly increased chromosome loss seen for the original Ch16 MC strain (i.e., two copies of *vtc4^+^*) at an incubation temperature of 36 °C was caused by the extra polyP. Although the strain lacking both copies of *vtc4^+^* showed a 76% reduction in Ch16 MC loss when compared to the strain harboring 2 copies of *vtc4^+^* (Figure 3b), the Ch16 MC loss of the strain unable to generate polyP was still significantly higher at 36 °C than at 30 °C (Figure 3b). Taken together, *vtc4^+^* copy number in a disomic strain increases polyP levels, which in turn augments chromosome segregation plasticity. Additionally, the high temperature has an independent effect on chromosome segregation (Figure 1c) as chromosome loss in the Ch16 MC strain without a *vtc4^+^* copy is still higher at 36 °C than 30 °C.

### 3.4. PolyP Impacts Kinetochore Function

To determine how polyP might affect chromosome segregation fidelity, we analyzed whether (i) a component of the conserved CCAN kinetochore subcomplex and/or (ii) microtubule function was affected by polyP levels. The rationale for this was our previous findings that Asp1 enzymatic activity is not only required for regulation of polyP generation but also modulates chromosome segregation fidelity via affecting the function of the kinetochore and microtubule spindle dynamics [42,43,44,56]. (i) In particular we have shown that components of the conserved multi-protein CCAN kinetochore subcomplex, which comprises an inner part of the kinetochore with a connection to centromeric chromatin, are modulated by 1-IPP levels [44,61]. Components of the *S. pombe* CCAN complex (named Mis6-Mal2-Sim4 complex in *S. pombe*), namely Mal2 (CENP-O ortholog) and Fta2 (CENP-P ortholog) which require each other for kinetochore localization, are targeted to the kinetochore in a 1-IPP-dependent manner [21,44,62] (Figure 4a). Reduced 1-IPP levels (which will lead to reduced cellular polyP) rescued the temperature-sensitive lethal phenotype of *fta2* and *mal2* mutant strains, *fta2-291* and *mal2-1*, respectively [44] (Figure 4a, right). Thus, a *fta2-291* mutant strain with a *vtc4^+^* deletion was made and the growth phenotype at various temperatures determined via a serial dilution patch test. Importantly, a strain with a *vtc4^+^* deletion had a similar inositol pyrophosphate profile as the wildtype strain (Appendix A).

Thus, we investigated whether absence of polyP affects the fitness of the *fta2-291* mutant strain. The *fta2-291* single mutant strain showed slightly reduced growth at 29 °C and was unable to grow at 31 °C (Figure 4b). In contrast, the *fta2-291 vtc4Δ* mutant strain grew better at all temperatures tested and was able to grow at 31 °C (Figure 4b). To determine whether absence of polyP affected the chromosome missegregation phenotype of *fta2-291* cells, we quantified the number of mitotic cells with unequally segregated chromatin, one of the main phenotypes of the *fta2-291* mutant strain [62] (Figure 4c). 65% of *fta2-291* mutant cells segregated their chromosomes unequally when incubated at 33 °C, but this was reduced significantly to 37% for the *fta2-291 vtc4Δ* mutant cells. Next, we tested whether the absence of polyP also affected the growth of other temperature-sensitive CCAN kinetochore mutant strains. We generated *vtc4∆* double mutants with *mis15-68* and *mis6-302*. Mis15 and Mis6 are components of the *S. pombe* CCAN kinetochore subcomplex [63,64,65,66]. As shown in Appendix A, the *mis15-68 vtc4Δ* mutant strain was able to grow better at higher temperatures than the single *mis15-68* mutant strain, while absence of polyP did not affect the growth of the *mis6-302* strain. Thus, the influence of polyP is not restricted to Fta2.

As absence of polyP rescued the temperature-sensitive growth phenotype of a *fta2-291* mutant strain, we tested whether higher than wild-type polyP levels would have the opposite effect and reduce the fitness of the *fta2-291* strain. Plasmid-borne expression of *vtc4^+^* via the thiamine-repressible *nmt1^+^* promoter increased polyP levels by 39% (Figure 5a).

Thus, high expression of Vtc4 in a wild-type strain is sufficient to significantly increase PolyP levels. Consequently, we transformed a control plasmid and a plasmid expressing *vtc4^+^* in wild-type and *fta2-291* strains and analyzed growth fitness through a serial dilution patch test. Low-level expression of *vtc4^+^* did not affect the growth fitness of either strain at 25 °C (Figure 5b) while high expression of *vtc4^+^* reduced the growth fitness of wild-type cells at all temperatures tested. However, the high expression of *vtc4^+^* in the *fta2-291* strain was lethal at all temperatures tested (Figure 5c). Thus, fitness of the *fta2-291* mutant strain is reduced by polyP in a dose-dependent manner.

Previously, we showed that kinetochore-targeting of Fta2 increased with decreasing 1-IPP levels [44] (Figure 5d). As decreasing 1-IPP levels will result in decreased polyP amounts, we tested whether kinetochore targeting of wild-type Fta2-GFP was affected by polyP amounts. Endogenous kinetochore-localized Fta2-GFP in interphase cells is seen as a single fluorescent dot as kinetochores are clustered adjacent to the spindle pole body [62,67] (Figure 5d). Cells expressing endogenous *fta2^+^-gfp* were transformed with either control plasmid or a plasmid expressing *vtc4^+^* via the *nmt1^+^* promoter. The expression of *vtc4^+^* did not reveal any difference in Fta2-GFP signal intensity at the kinetochores (Figure 5e; images and quantification). Thus, *vtc4^+^* overexpression leads to higher levels of PolyP which exacerbates the temperature-dependent growth phenotype of the *fta2-291* strain through a mechanism independent of Fta2 targeting the kinetochore.

(ii) Our finding that polyP modulates kinetochore function raised the question of whether it also affected another mitotic machinery, namely the spindle, as this is also modulated by 1-IPPs. Thus, we tested whether polyP affected microtubule function by using two assays. First, we tested whether microtubule stability was altered in the *vtc4Δ* (polyP-deficient) strain. As expected, the presence of the microtubule-poison thiabendazole (TBZ) impacted the growth of the *mal3∆* strain (Appendix A), as *mal3^+^* codes for a member of the microtubule-plus-end EB1 family which controls microtubule dynamics [23,68]. However, growth on TBZ of the *vtc4Δ* strain was comparable to that of the wild-type strain (Appendix A), demonstrating that loss of polyP has no impact on TBZ sensitivity of the strain. This is in contrast to the *asp1∆* and *asp1^D333A^* strains, which are unable to generate 1-IPPs [43,55,69]. Second, we determined that alteration of the microtubule-mitochondria crosstalk in the *vtc4∆* strain as association of mitochondria with interphase microtubules determines their tubular structure and loss of Asp1 kinase activity (and thus the ability to generate polyP) disrupts this interaction (Appendix A) [19,43,70,71]. It is well known that polyP is present in mammalian mitochondria and has an impact on the bioenergetics of these organelles, but whether such a connection exists in fission yeast is not known [72]. The data below indicate that this is probably not the case. Live-cell microscopic analysis of mitochondrial morphology using the tagged mitochondrial Cox-RFP protein revealed that the Cox4-RFP signal and thus mitochondrial distribution in the *vtc4∆* strain was similar to that of the wild-type (Appendix A). This is in contrast to the phenotype of mitochondria in the *asp1^D333A^* strain, which cannot generate 1-IPPs and polyP (Appendix A). We conclude that it is unlikely that polyP has an impact on spindle microtubules.

### 3.5. Nucleolin Plays a Role in CCAN Kinetochore Complex Function

To decipher how polyP might affect chromosome transmission fidelity, we next analyzed the possible impact of *S. pombe* nucleolin Gar2 [73]. The rationale for this was a previous finding that *S. cerevisiae* Nsr1, which is a functional ortholog of *S. pombe* Gar2, shows polyP-mediated post-translational modification namely polyphosphorylation (Figure 6a) [39,73]. This modification has functional consequences [39]. At present it is unclear whether such a post-translational modification is also found for *S. pombe* Gar2, but the polyP-relevant PASK domain is highly conserved. *S. pombe* Gar2 and *S. cerevisiae* Nsr1 belong to the nucleolin family, predominantly present in the nucleolus and involved in numerous cellular processes, the best studied being its role in ribosome biogenesis [74]. In mitosis, nucleolin at the chromosome periphery plays a role in microtubule-kinetochore attachment and fission yeast Gar2 impacts centromeric chromatin structure [75,76,77]. We revisited the localization of Gar2 and found, as previously described, that Gar2-GFP was localized to the nucleolus when cells were grown in liquid culture (Figure 6b) [76]. However, when Gar2-GFP cells were grown on solid media prior to microscopic analysis, we found additional small Gar2 fluorescent dots in addition to the nucleolar signal in 54% of cells (Figure 6b, right panel). These extra Gar2 dots were not in proximity to the centromere (LacI-GFP LacO-repeat:lys1^+^) and only in 7.6% of cells close to the spindle pole body (Sad1-mCherry) (Appendix A) [78].

To test whether Gar2 affects CCAN kinetochore complex components Fta2 and Mal2 (CENP-O ortholog), we generated double mutant strains: *fta2-291* and *mal2-1* strains with a deletion of the *gar2^+^* ORF (*gar2Δ)*. *gar2Δ* strains showed a slightly reduced growth at 25 °C when compared to wild-type control (Figure 6c and Appendix A). At higher temperatures, *gar2Δ* partially rescued the *fta2-291* temperature-sensitive growth phenotype at 32 °C, similarly to what we observed when we deleted *vtc4^+^* (Figure 4b). Next, we tested whether the deletion of *gar2^+^* could also rescue the growth sensitive phenotype of the *mal2-1* mutant strain. Indeed, we found that the temperature-caused non-growth phenotype of the single *mal2-1* mutant was not seen for the *gar2Δ mal2-1* double mutant strain (Appendix A). Thus, Gar2 plays a role in the function of the CCAN kinetochore complex.

Next, we determined whether the lethal phenotype observed for the *fta2-291* mutant strain in the presence of higher than wild-type polyP levels (i.e., *vtc4*^+^ overexpression) was suppressed in the absence of *gar2^+^*. Therefore, we transformed parental strains *gar2Δ* and *fta2-291* as well as a *fta2-291 gar2Δ* double mutant strain with a plasmid harboring *vtc4^+^* under the control of the *nmt1^+^* promoter. We grew the strains at 29 °C, a temperature at which there is little difference in growth fitness between all three strains (Figure 6c). Under low-level expression of plasmid-borne *vtc4^+^*, all transformants grew similarly to the cells transformed with the control plasmid (Figure 6d). However, all strains harboring the *vtc4^+^* plasmid grown on media that allows high *vtc4^+^* expression showed growth defects. Interestingly, whereas the *fta2-291* single mutant strain failed to thrive, the *fta2-291 gar2Δ* double mutant strain showed a marked increase in growth fitness (Figure 6d). Taken together, we conclude that Gar2 is one of the proteins required to mediate the full growth impairment resulting from *vtc4^+^* overexpression in a strain with a deficient kinetochore function.

## 4. Discussion

In the present work, we used an *S. pombe* strain with a small segmental aneuploidy to determine whether moderate changes in temperature affect chromosome transmission fidelity and found this to be the case. To assess whether a gene dosage effect contributed to the observed chromosome loss, we studied the impact of inorganic polyphosphate on chromosome transmission accuracy. The rationale behind this was our previous observation that specific inositol pyrophosphates made by the *S. pombe* PPIP5K enzyme Asp1 modulate chromosome segregation fidelity and are also essential for polyP synthesis [19,42,44]. In *S. pombe*, polyP is generated by the 3-component VTC complex and *vtc4*^+^, which is present both on chromosome III and the MC Ch16, encodes the polyphosphate synthase. Intriguingly, we found that the extra *vtc4*^+^ copy led to 30% higher than wild-type polyP levels and that chromosome transmission fidelity was inversely correlated with polyP levels. Thus, the presence of two *vtc4*^+^ copies leads to increased generation of polyP levels even though polyP is generated by a protein complex. In *S. cerevisiae,* where polyP generation is also modulated by the PPIP5K member, the subunit stoichiometry of one of the two existing VTC complexes has been determined to be 1:1:3 (Vtc4/Vtc3/Vtc1) [53,54,79,80]. Our work defines a hitherto unknown role of polyP in genome plasticity via gene dosage. As polyphosphate is involved in stress response and cellular polyP levels vary in response to specific conditions, it is possible that in such scenarios, chromosome transmission fidelity is also altered [35,81,82]. Our analysis also shows that an extrinsic variable namely temperature and an intrinsic variable namely polyP contribute independently to chromosome transmission fidelity.

### 4.1. Moderate Growth Temperature Changes and Chromosome Transmission Fidelity

In nature, yeasts experience constant temperature changes and are usually adapted to drive within a temperature range. However, at temperatures above 36 °C, yeasts such as *S. cerevisiae* will activate the heat shock response to deal with heat stress [83]. Furthermore, at suboptimal lower growth temperatures gene expression patterns are changed dramatically, including upregulation of stress response genes [84]. Thus, we analyzed chromosome segregation fidelity of the Ch16 MC strain at moderate growth temperatures but still found a correlation between temperature and chromosome loss. Interestingly such differences do not appear to be caused in aneuploid strains only. In a recent study, wild-type, haploid *S. pombe* euploids mitotic cells were analyzed microscopically at different temperatures and showed a gradual increase in chromosome segregation defects when cultures were incubated at temperatures higher than 25 °C [85]. Despite the differences in strains and assays used, the authors also found a slight increase in chromosome segregation defects at 30 °C when compared to 25 °C [85]. In both systems, a dramatic increase in chromosome loss was observed at the highest temperature tested, 40 °C [85] and 36 °C (the present study). Higher temperature is a well-documented stressor in many fungal species [86,87], and genome plasticity may provide a mechanism for thermal adaptation.

### 4.2. PolyP Dosage and Chromosome Transmission Fidelity

The Ch16 MC strain has been used in numerous *S. pombe* studies as an assay for genome instability, which can be scored easily because the MC is lost at a higher frequency than an endogenous chromosome in the haploid strain [16]. Our data suggest that the Ch16 MC assay system is an involuntarily sensitized system to study chromosome loss due to the extra copy of *vtc4*^+^ present on the Ch16 MC. An increased rate of chromosome mis-segregation is directly related to the cellular polyP levels, demonstrating that polyP levels contribute to chromosomal instability (CIN). Aneuploidy leading to CIN is frequently observed in cancer and has an impact on progression of the disease [88]. PolyP is also present in mammalian cells and although the amount, chain length and its localization in the cell varies depending on the cell type analyzed, high concentration can also be found in the nucleus [30,33,58,89]. Intriguingly, different cancer cell lines treated with the chemotherapy drug cisplatin accumulate high levels of polyP mainly in the nucleolus which correlates with caspase-mediated apoptosis [90]. As treatment with cisplatin can lead to aneuploidy/chromosomal abnormalities, it is possible that polyP also has a role in CIN in mammalian cells [91]. Our present work reveals that Vtc4-dependent levels of polyP play a role in segregation fidelity, namely, increased levels of polyP increase chromosome loss, while decreased levels of polyP decrease it. How might polyP regulate chromosome segregation?

Firstly, we identified a pool of polyP in nuclear *S. pombe* fractions, similar to what has been shown in other species [33,39,89], allowing polyP to have access to chromatin and the multiple players required for chromosome transmission. In line with this, in *S. pombe vtc4∆* cells, transcriptional alteration of a number of genes has been observed [92], and polyP modulates bacterial heterochromatin formation [93,94], suggesting a broad role in chromatin regulation.

### 4.3. PolyP and the Kinetochore

Secondly, we identified a target for polyP in the nucleus, namely the *S. pombe* Fta2 (CENP-P) protein, a component of the highly conserved CCAN kinetochore subcomplex [62,65,95]. The multicomponent CCAN complex is required for and implements a robust interface with centromeric chromatin [61]. Centromeric chromatin is specialized chromatin regulated epigenetically and defined by the presence of the special histone H3 variant CENP-A [96]. Components of CCAN such as Fta2 impact on centromeric chromatin specifically that of the central core region where CENP-A nucleosomes are found and on which the kinetochore complex is built [62,97]. We now find, that the absence of Vtc4-dependent polyP rescued the temperature-dependent reduction in growth and the chromosome segregation defects of the mutant *fta2-291* strain, while higher than wild-type levels of polyP had the opposite effect. Such a rescue in growth was not restricted to the *fta2-291* mutant but also observed for another mutant component of the CCAN.

Genetic analysis points to a possible route how polyP might impact kinetochore function as the polyP impact occurs- at least partially- via the nucleolin Gar2. Gar2 is the functional homolog of the *S. cerevisiae* Nsr1 protein [73]. Nsr1 is post-translationally modified by polyP resulting in polyphosphorylation and this modification has a negative impact on the interaction of Nsr1 with Top1 (topoisomerase I) [39]. As (i) the absence of either *gar2^+^* or *vtc4*^+^ have a similar effect on the survival of the *fta2-291* strain at semi-permissive temperatures and (ii) plasmid-borne overexpression of *vtc4^+^* (higher levels of polyP) which is lethal for the *fta2-291* strain can be rescued partially when *gar2^+^* has been deleted, it is feasible, that one way in which polyP alters chromosome transmission fidelity is via Gar2 modulation which then impacts on kinetochore function. Importantly, it has been shown previously that Gar2 plays a role in epigenetic regulation of the central core region of the *S. pombe* centromere chromatin and is a mass spectrometry-identified interaction partner of Ccp1 [76]. Ccp1, belongs to the family of nucleosome assembly proteins, and is associated with centromere chromatin in interphase but not mitosis and antagonizes loading of CENP-A [76,98]. Thus, we propose that one of the avenues taken by polyP to modulate chromosome transmission fidelity is to modulate the centromere chromatin-CCAN crosstalk.

### 4.4. PolyP and Asp1-Modulated Chromosome Segregation

We have previously shown that Asp1-made 1-IPPs have an important impact on chromosome segregation fidelity as Asp1 kinase activity regulates (i) microtubule dynamics and spindle formation and function [42,43,56] and (ii) kinetochore targeting of CCAN components [44]. In this work, we show that polyP generated by the VTC complex, which is also regulated by PPIP5K kinase activity, has a dosage-dependent function in modulating chromosome transmission fidelity. Thus, is the Asp1-mediated impact on a functional mitosis caused indirectly by its regulation of polyP synthesis? Our analysis does not support such a scenario. Firstly, 1-IPPs are required in a dose-dependent manner for microtubule dynamics, and thus a strain unable to synthesize 1-IPPs such as the *asp1^D333A^* strain is highly sensitive to microtubule poisons and has defects in the microtubule-mitochondria crosstalk as interphase microtubules regulate mitochondrial distribution [43,56,71]. However, we show here that cells without *vtc4^+^* have no increased sensitivity to the microtubule poison TZB and a wild-type like distribution of mitochondria. Secondly, although both lower 1-IPP and polyP levels rescue the growth defects of a mutant Fta2 strain, 1-IPPs control targeting of CCAN components to the kinetochore while polyP does not [44]. Thirdly, an increase in higher than wild-type 1-IPP levels increases transmission fidelity of the Ch16 MC [42,43]. Thus, we suggest that the regulation of kinetochore function through 1-IPPs and polyP is elastic, allowing for inputs derived from diverse environmental and/or cellular demands to regulate mitotic chromosome transmission fidelity appropriately.

## 5. Conclusions

In the fission yeast *S. pombe*, a new function for inorganic polyphosphate has been discovered, namely a role in chromosome transmission fidelity. Genetic evidence points to polyP modulation of the conserved CCAN kinetochore complex.

## Figures and Tables

**Figure 1 biomolecules-15-01331-f001:**
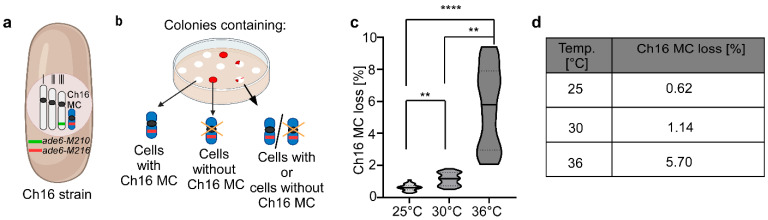
Ch16 MC loss rate is temperature-dependent. (**a**) Schematic representation of the *S. pombe* Ch16 strain, containing the three endogenous chromosomes I–III and a 530 kb mini-chromosome (Ch16 MC), a derivative of chromosome III [16]. Chromosome III carries the *ade6-M210* marker (green bar), while the Ch16 MC carries *ade6-M216* (red bar). Cells that contain both *ade6* alleles are adenine prototrophs, whereas loss of the Ch16 MC results in adenine auxotrophy (**b**) Diagrammatic representation of the Ch16 MC loss assay. White colonies consist of cells that have the Ch16 MC, red colonies consist of cells that have lost the Ch16 MC. White colonies with red sectors (sectored colonies) are colonies where the red sectors contain cells which have lost the Ch16 MC and thus they and their progeny turned red. Only white and sectored colonies were counted for the analysis. (**c**) Ch16 MC loss at different temperatures was quantified as the percentage of sectored colonies out of the total colony number (sectored + white), as shown on the *y*-axis. For each temperature data from 2–3 independent experiments were combined into a single graph. Colonies analyzed per temperature: total colonies/sectored colonies; 29,354/181 (25 °C), 23,777/269 (30 °C), 5663/296 (36 °C). Statistical analysis was performed using GraphPad Prism v.10.4.1. Normality was assessed using a Shapiro–Wilk test. As data were not normally distributed, a Kruskal–Wallis test followed by Dunn’s multiple comparisons test was applied. *p*-values are adjusted for multiple comparisons. Significance levels: *p*  <  0.01 (**), *p * <  0.0001 (****). Data represent median values ± interquartile range (IQR). (**d**) Table showing mean loss rates of the Ch16 MC with the data derived from Figure 1c.

**Figure 2 biomolecules-15-01331-f002:**
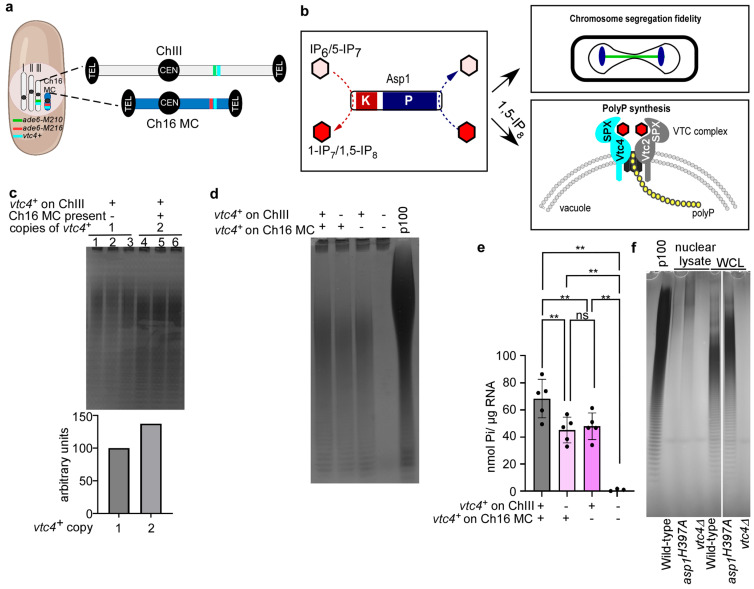
The presence of two copies of *vtc4^+^* results in increased polyphosphate (polyP) levels. (**a**) Diagrammatic representation of the *S. pombe* Ch16 MC strain containing endogenous chromosomes I–III and the Ch16 MC. Drawing is not to scale. The *vtc4^+^* gene is present in two copies: on chromosome III and on the Ch16 MC. Chromosomes are not to scale. Telomeres (TEL) and centromeres (CEN) are indicated. The position of the *vtc4^+^* genes is shown in turquoise. (**b**) Schematic representation of inositol pyrophosphate (IPP) synthesis by the bifunctional enzyme Asp1 and its established role in chromosome segregation fidelity and polyP generation [19,42,44]. The *N*-terminal kinase domain (K, red) phosphorylates IP_6_ and 5-IP_7_ at the 1-position of the inositol ring (indicated by a pink-to-red transition), while the C-terminal phosphatase domain (P) removes the diphosphate group from the same position. VTC = vacuolar transporter chaperone complex components Vtc1 (gray), Vtc4 (blue) and Vtc1 (black). SPX = IPP binding domain of Vtc2 and Vtc4 as shown for the *S. cerevisiae* complex [53,54]. (**c**) Qualitative analysis of polyP by native PAGE followed by toluidine blue staining. Equal amounts of total RNA (20 μg) were loaded per lane. Lanes 1–3 show technical replicates of a haploid wild-type strain, while lanes 4–6 show technical replicates of the Ch16 MC-containing strain. Densitometric analysis was conducted using ImageJ by integrating the area under the intensity profile curve of the gel lanes, representing the total gray value intensity of the bands. The Ch16 MC-containing strain showed a 37% increase in polyP compared to the wild-type strain. (**d**) Toluidine blue-stained native PAGE gel showing polyP extracted from *S. pombe* Ch16 MC containing strains that have either two, one or no copies of the *vtc4^+^* ORF. The two strains with only one copy were generated by retaining *vtc4^+^* on chromosome III and deleting it from the Ch16 MC, or vice versa. Equal amounts of RNA (20 μg) were loaded per lane. The presence of *vtc4^+^* and the genomic position are indicated by + (present) or - (deleted). P100 is a polyP standard with an average length of 100. (**e**) Quantitative analysis of polyP levels from the strains shown in Figure 2d. Samples were treated with recombinant *S. cerevisiae* Ddp1 and Ppx1 polyphosphatases, and release of inorganic phosphate (Pi) was measured using a malachite green colorimetric assay. Statistical analysis was performed using GraphPad Prism v.10.4.1. A one-way ANOVA followed by Tukey’s multiple comparisons test was applied. *p*-values are adjusted for multiple comparisons. Significance levels: *p*  <  0.01 (**). ns, not significant. Shown are the data of 5 biological replicates per strain. Data represent mean values ± SD. (**f**) Qualitative assessment of nuclear polyP lysates of the indicated strains. Loading from left to right: *asp1^+^ vtc4*^+^ (wild-type strain), *asp1^H397A^ vtc4*^+^ (*asp1* variant that generates higher than wild-type 1-IPP and thus polyP amounts) [19,43,55], *asp1^+^ vtc4Δ* (*vtc4^+^* deletion strain) by native PAGE and DAPI staining. Shown are nuclear lysates and whole cell lysate (WCL).

**Figure 3 biomolecules-15-01331-f003:**
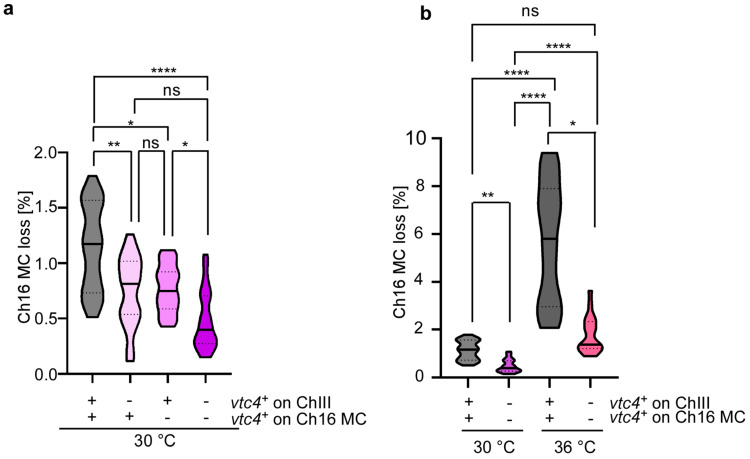
Ch16 MC loss is modulated independently by temperature and polyP levels. (**a**) Violin plots showing Ch16 MC loss rates [%] at 30 °C in *S. pombe* strains carrying *vtc4^+^* on both chromosome III and the Ch16 MC, or with *vtc4^+^* deleted at one or both loci. The *Y*-axis represents the percentage of sectored colonies. The *X*-axis displays the four genotypes: (++) *vtc4^+^* present on both chromosome III and Ch16 MC; (-+) *vtc4^+^* deleted on chromosome III; (+-) *vtc4^+^* deleted on Ch16 MC; (--) *vtc4^+^* deleted at both loci. Total and sectored colony counts: (++) 23,777/269; (-+) 22,758/170; (+-) 26,201/202; (--) 24,069/116. The experiment for each strain was independently repeated two to three times, and data from all replicates were pooled. Statistical analysis was performed using GraphPad Prism v.10.4.1. Normality was assessed using a Shapiro–Wilk test. As data were not normally distributed, a Kruskal–Wallis test followed by Dunn’s multiple comparisons test was applied. Outliers were identified and excluded using the ROUT method (Q = 5%). One outlier was removed from 23 data points in the (--) strain. *p*-values are adjusted for multiple comparisons. Significance levels: *p*  <  0.05 (*), *p*  <  0.01 (**), *p * <  0.0001 (****). ns, not significant. Data represent median values ± IQR. (**b**) Violin plots showing Ch16 MC loss rates [%] at 30 °C and 36 °C. Strains either have two *vtc4^+^* genes (++) or no *vtc4^+^* copy (--). Data for strain (++) at 30 °C and 36° C were shown in Figure 1c and Figure 3a. Data for strain (--) at 30 °C were also presented in Figure 3a. At 36 °C, 358/20,076 colonies of the (--) strain were sectored. Statistical analysis was carried out as in Figure 3a. Outliers were removed using the ROUT method (Q = 5%): one from 23 data points in the (--) strain at 30 °C, and one from 21 data points in the (-) strain at 36 °C.

**Figure 4 biomolecules-15-01331-f004:**
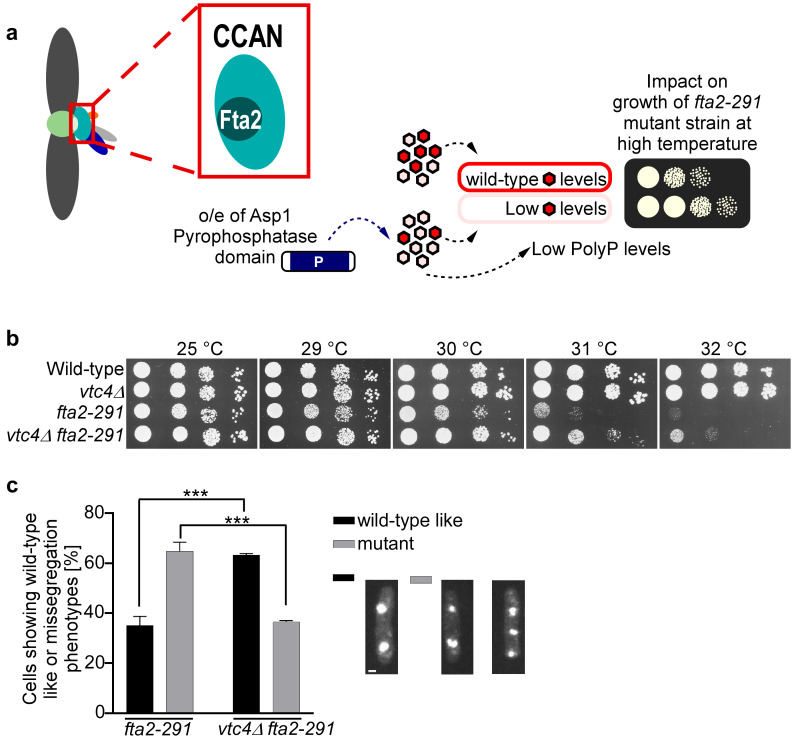
Genetic ablation of the polyP polymerase Vtc4 partially rescues the temperature-sensitive growth phenotype of the CCAN kinetochore mutant *fta2-291* strain. (**a**) Schematic representation of the impact of Asp1-derived 1-IPPs on Fta2 function. Left, Fta2 is part of the *S. pombe* CCAN kinetochore complex (cyan) [62]. Centromere chromatin (light green). The other colors represent different kinetochore subcomplexes. Right, plasmid-borne expression of the Asp1 pyrophosphatase domain (blue) decreases the levels of 1-IPPs (5-IP_7_ in rose and 1,5-IP_8_ in red) [43], which partially rescues the temperature-sensitive growth phenotype of the *vtc4^+^ fta2-291* mutant strain by re-localizing the mutant Fta2 protein to the kinetochore [44]. o/e = overexpression. (**b**) Serial dilution patch test (10^4^ to 10^1^ cells) of the indicated strains grown for 4–5 days on YE5S plates at the specified temperatures. (**c**) Left: Quantification of chromosome missegregation frequencies of indicated strains. Cells were precultured at 25 °C, shifted to 33 °C for 6 h, fixed, stained with DAPI and segregation of chromatin in anaphase cells scored microscopically. Statistical analysis was performed using GraphPad Prism v.10.4.1. A two-way ANOVA followed by Šídák’s multiple comparisons test was applied. *p*-values are adjusted for multiple comparisons. Significance levels: *p*  <  0.001 (***). Data represent two independent experiments with mean values ± SD. Number of anaphase B cells analyzed: *vtc4^+^ fta2-291* = 205, 342; *vtc4Δ fta2-291* = 216, 367. Right: Representative images of mitotic cells exhibiting either equal chromatin segregation (black bar) or segregation defects (gray bar). Scale bar: 1 µm.

**Figure 5 biomolecules-15-01331-f005:**
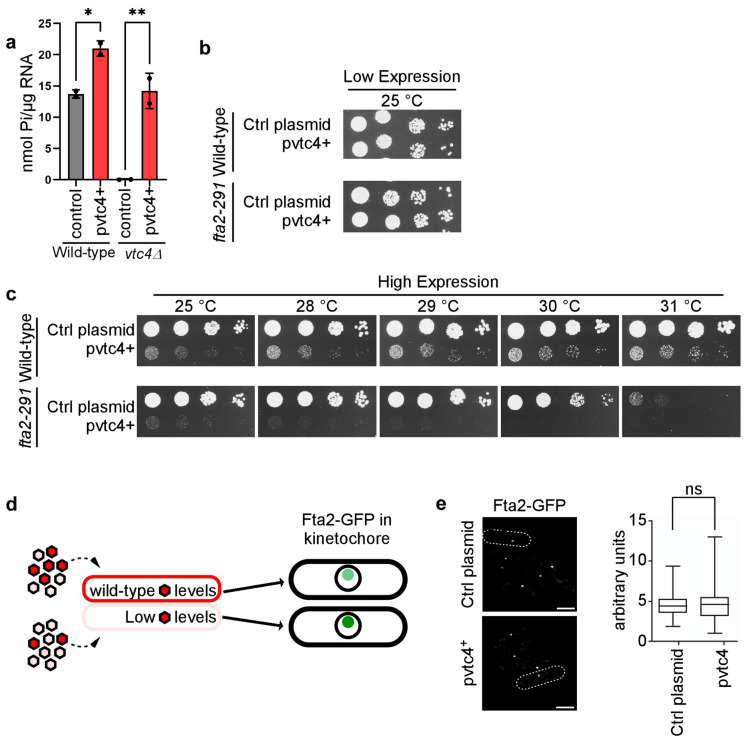
Elevated polyP levels impair viability of the kinetochore mutant *fta2-291.* (**a**) Quantitative analysis of polyP levels in *S. pombe* wild-type and *vtc4Δ* strains transformed with either a control plasmid or a plasmid expressing *vtc4^+^* under the control of the thiamine-repressible *nmt1^+^* promoter Transformants were grown in media without thiamine, which leads to high-level expression of *vtc4^+^*. PolyP levels were quantified by enzymatic degradation using Ddp1 and Ppx1, followed by a malachite green assay measuring the released inorganic phosphate (Pi). The bar graph shows two biological replicates. Statistical analysis was performed in GraphPad Prism v.10.4.1 using unpaired two-tailed Student’s t-tests. Significance levels: *p*  <  0.05 (*), *p*  <  0.01 (**). ns, not significant. Data represent mean values ± SD. (**b**) Serial dilution patch test (10^4^ to 10^1^ cells) of *S. pombe* wild-type and *fta2-291* mutant strains transformed with either a control plasmid or a plasmid expressing *vtc4^+^*. Cells were grown under plasmid selective conditions and in the presence of thiamine to repress the *nmt1^+^* promoter (low expression). (**c**) Serial dilution patch test as in (**b**), but cells were grown in thiamine-free medium for high expression of *vtc4^+^*. Plates were incubated at the indicated temperatures for 4–5 days. (**d**) Diagrammatic representation of Fta2-GFP kinetochore targeting under different 1-IPP conditions. 5-IP_7_ shown in rose and 1,5-IP_8_ in red. Fta2-GFP kinetochore signal intensity (light to dark green) inversely correlates with 1-IPP levels [44]. (**e**) Left: Live-cell images of *S. pombe* cells expressing Fta2-GFP from the endogenous promoter, transformed with either a control plasmid or a plasmid overexpressing *vtc4^+^* via the *nmt1^+^* promoter. Cells were grown at 30 °C in medium without thiamine. Scale bars: 5 µm. Dashed lines mark the cell. Right: quantification of Fta2-GFP fluorescence signals of the indicated transformants. Data represent mean values ± SEM: control plasmid = 4.57 ± 0.15 AU; pvtc4^+^ = 4.51 ± 0.16 AU, *p* = 0.78. Number of kinetochore signals analyzed: control plasmid, *n* = 111; pvtc4^+^, *n* = 135. An unpaired two-tailed Student’s *t*-test with Welch’s correction was used.

**Figure 6 biomolecules-15-01331-f006:**
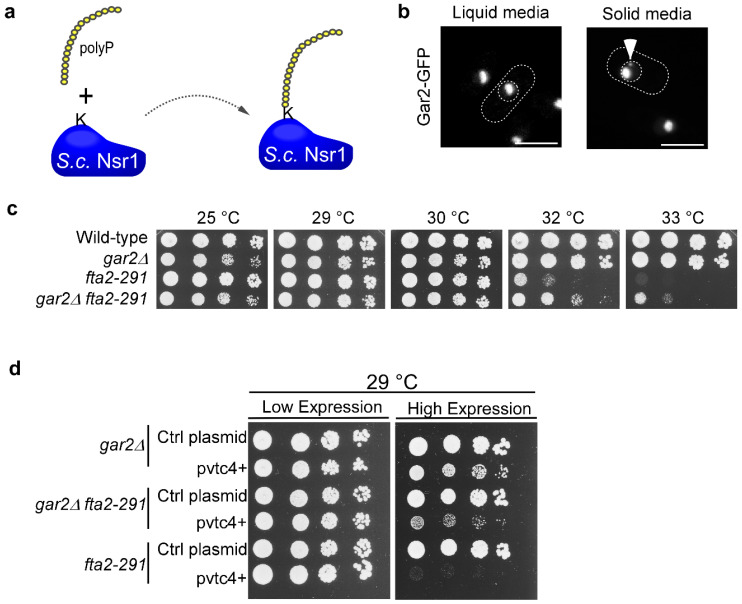
Genetic interaction between *fta2-291* and *gar2^+^*, encoding Nucleolin. (**a**) Schematic illustration to show the posttranslational modification of *S. cerevisiae* protein Nsr1, a functional homolog of *S. pombe* Gar2 [39,73]. (**b**) *S. pombe* cells endogenously expressing Gar2-GFP were grown at 30 °C in liquid culture or on solid medium and subsequently imaged by live-cell fluorescence microscopy. Dashed lines mark the cell (cylindrical) and nuclear (round) boundaries. The arrow highlights an additional Gar2-GFP signal observed under solid media conditions. Scale bar: 5 µm. (**c**) Serial dilution patch test (10^4^–10^1^ cells) of the indicated strains grown for 4-5 days on YE5S plates at the shown temperatures. (**d**) Serial dilution patch test (10^4^–10^1^ cells) of *S. pombe* strains (*gar2Δ*, *fta2-291*, and *fta2-291 gar2Δ*) transformed with either a control plasmid or a plasmid expressing *vtc4^+^* under the control of the thiamine-repressible *nmt1^+^* promoter. Cells were incubated at 29 °C for 4–5 days on plasmid-selective minimal medium containing thiamine (low expression) or lacking thiamine (high expression).

## Data Availability

Data are contained within the article or Appendix A.

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
