# Peer review of "Inorganic Polyphosphate Modulates Chromosome Transmission Fidelity in the Fission Yeast Schizosaccharomyces pombe"

_biomolecules, 2025, doi:10.3390/biom15091331_

Round 1
Reviewer 1 Report
Comments and Suggestions for Authors
In this manuscript, the authors report a role for vtc4+ gene dosage on chromosome transmission fidelity and they nicely show that these effects correlate with increased polyP levels. Based on a series of genetic tests using mutant strains and overexpression studies, the authors show that polyP affects kinetochore function and their findings regarding Gar2 (the homolog of S. cerevisiae Nsr1 in fission yeast) raise the exciting possibility that Gar2 function might be modulated by polyP-mediated polyphosphorylated in S. pombe as it is in S. cerevisiae. Overall, this is an interesting study that provides new information about chromosome transmission fidelity and the impact of polyP.
Specific comments:
1. According to the Figure legends, the bar graphs in figures 2e and 5a show one representative experiment out of two, each with two technical replicates. For the quantitative measurements (with statistical analysis) the authors should show the data of the mean values (±SD) for two biological replicates (not for two technical replicates).
2. The authors report that polyP in the nucleus represents 0.01% of total polyP in cells. More details are needed to understand how the authors arrive at this conclusion. For example, saying that equal volumes of nuclear and whole cell lysates were loaded (line 172) does not mean much without knowing to how many cells (e.g. as OD units) it corresponds to in each case (i.e. what were the final volumes for nuclear extract, what was the volume for WCL from 50 OD units?) Also, the experiments used to quantify polyP levels in nuclear fractions should be mentioned in the text rather than in the legend to Supplemental figure 2.
Minor comments and typos:
- The model in Fig. 3c is difficult to interpret without knowing what the triangle represents. For example, if one assumes that the pointed end reflects “low”, shouldn’t the triangle for MC loss be flipped horizontally (e.g. high temp leading to high MC loss)? In any event, panel 3c could be removed.
- Figure legends: Bold font is used for the heading of panels (a), but not for the other panels (or for the overall Figure heading). Is this intentional?
- 2: please mention p100 in the legend (what is it?)
- line 290: Fig. 2c legend refers to lanes 1-3 and 4-6, but the lanes are not labeled as such in the figure itself.
- line 294: the percentage increase (37%) is not shown in panel 2c. I suggest to include the relative percentages of polyP below the gel image.
- line 123: please clarify “for strains, the procedure was the same…” as to what ?
- line 139: what does “probes” refer to?
- line 235: …colony color assay of cells that carry two ade6 alleles instead of color colony assays which carry two ade6 alleles?
- line 413; remove (i)
- line 416: remove Asp1
- line 447: “only determining” sounds awkward
- line 453 and 455: the words ‘single’ and ‘double’ are superfluous
- line 499: please fix the sentence ….with plasmids expressing either control plasmid or….
- line 512: insert vtc4∆ to read (…growth on TBZ of the vtc4∆ strain was comparable…)
- line 576: Fig. 6c instead of Fig 5c
- line 650: I’m not sure if there is a benefit to mention transcriptional alterations in vtc4∆ cells (ref 86) without saying anything of whether those changes might be related (or not) to the phenotypes observed in the present study.
- line 620: …at temperatures higher than 25°C (instead of at higher temperatures than 25°C)
- line 709: delete either ‘role’ or ‘function’
Reviewer 2 Report
Comments and Suggestions for Authors
The fission yeast Schizosaccharomyces pombe is a premier model for studying chromosome segregation as its centromeres possess the typical repetitive structure of higher eukaryotic counterparts, and the network of kinetochore proteins (e.g. CCAN) identified in vertebrates is also conserved. In this manuscript, Bollé and colleagues show that segregation of chromosomes in S. pombe is affected by temperature and modulated by the concentration of inorganic polyphosphate polymers (polyP), which in turn affects the function of conserved CCAN proteins Fta2/CENP-P and Mal2/CENP-O. Overall, the paper is well written and the presented experimental evidence, which include genetics, biochemistry and live cell microscopy, is very convincing. I really enjoyed reading it and I have only a few suggestions that can improve this already fine study.
Introduction (lines 58-59):
Regarding fission yeast diploids, the authors should also cite PMID: 28199302, which employs laboratory-generated diploid S. pombe cells to study uniparental disomy, a hallmark of cancer.
Fig. 1:
Showing that another mini-chromosome that does not contain vtc4+, perhaps a circular one generated by Yanagida or Clarke labs, is not drastically affected by temperature would reinforce the claim that polyP is the main driver of mis-segregation at high temperature.
Fig. 2a:
The schematic is not in scale and suggests that ade6+ is closer to the right telomere rather than centromere 3, which is in fact the opposite.
Fig. 4 b,c:
Showing similar results with another mutant involved in polyphosphate biosynthetic process, such as vtc1∆ or vtc2∆, would further validate the data obtained with vtc4∆. On the other hand, it would be nice to see no rescue by vtc4∆ of widely used kinetochore thermosensitive mutants (e.g. mis6-302, mis12-537).
Fig. 5:
Similar rationale as in the comments for Fig. 4.
Fig. 6b and Supp. Fig. 5:
Can authors provide quantification of these configurations?
Discussion:
Since it is unclear how polyP affects Fta2/CENP-P function, an exciting possibility could be that Fta2 (Ctf19 in budding yeast) might aberrantly recruit cohesin at the central core upon high levels of polyP (see Hinshaw et al., Cell 2017 PMID: 28938124). Can the authors comment on this?
